

# *bmd*: an R package for benchmark dose estimation

Signe M. Jensen[1], Felix M. Kluxen[2], Jens C. Streibig[1], Nina Cedergreen[3] and Christian Ritz[4]

[1] Department of Plant and Environmental Sciences, University of Copenhagen, Taastrup, Denmark
[2] ADAMA Deutschland GmbH, Cologne, Germany
[3] Department of Plant and Environmental Sciences, University of Copenhagen, Frederiksberg C, Denmark
[4] Department of Nutrition, Exercise and Sports, University of Copenhagen, Frederiksberg C, Denmark

## ABSTRACT

The benchmark dose (BMD) methodology is used to derive a hazard characterization measure for risk assessment in toxicology or ecotoxicology. The present paper's objective is to introduce the **R** extension package *bmd,* which facilitates the estimation of BMD and the benchmark dose lower limit for a wide range of dose-response models via the popular package *drc*. It allows using the most current statistical methods for BMD estimation, including model averaging. The package *bmd* can be used for BMD estimation for binomial, continuous, and count data in a simple set up or from complex hierarchical designs and is introduced using four examples. While there are other stand-alone software solutions available to estimate BMDs, the package *bmd* facilitates easy estimation within the established and flexible statistical environment **R**. It allows the rapid implementation of available, novel, and future statistical methods and the integration of other statistical analyses.

## INTRODUCTION

Risk assessment in human toxicology and ecotoxicology quantifies the relationship between the exposure to a chemical (or other stressor) and a hazard characterization measure of the chemical. If the exposure dose/concentration is estimated to be less than the dose/concentration estimated to have adverse effects (the point of departure (POD)) then the risk associated with exposure to the chemical agent is considered acceptable. In practice, the POD is often modified by safety or uncertainty factors to account for inter-species differences and inter- and intra-individual differences in the target population. This modified value is usually called the reference dose/concentration or limit value, depending on the context. The two main approaches for deriving PODs are the No Observed Adverse Effect Level (NOAEL) and the Benchmark Dose (BMD) methodology. Note that for concentrations also NOAEC (No Observed Adverse Effect Concentration) and less commonly BMC (Benchmark Concentration) are used, as is NOEL, (No Observed Effect Level), NOED (No Observed Effect Dose) or NOEC, (No Observed Effect Concentration), in the context where any effect is considered adverse. Whether level, dose, or concentration is used depends on how the test organism/system is exposed to the chemical agent, but in

Corresponding author
Signe M. Jensen, smj@plen.ku.dk

the context of developing methods for benchmark dose estimation the term dose will be used to quantify exposure.

The NOAEL is the highest dose that does not lead to an observable adverse change in the investigated response. Hypothesis testing usually complements the Lowest Adverse Effect Level (LOAEL) determination, i.e., the next dose level above the NOAEL. The NOAEL will be an experimentally tested dose and it depends strongly on the dose spacing. Poor experimental design, high variation, and low statistical power lead to relatively higher NOAELs, which is problematic from a conservative point of view and can only be mitigated by toxicological weight-of-evidence assessment, considering multiple studies or endpoints within a single study or by means of additional safety factors.

The BMD approach aims to reduce some of these issues associated with NOAEL derivation and relies on dose–response modelling. It is, therefore, less dependent on the tested doses. The BMD methodology was initially introduced by Crump (*Crump, 1984*) for binomial response data, but has subsequently been extended with definitions for continuous response data (*Gaylor & Slikker, 1990*; *Crump, 1995*; *Budtz-Jørgensen, Keiding & Grandjean, 2001*). The BMD methodology is now recommended by OECD and other regulatory authorities (*Davis, Gift & Zhao, 2011*; *Organisation for Economic Co-operation and Development , 2012a*; *Hardy et al., 2017*). The BMD is defined as the dose associated with a predefined small change in the response, the benchmark response (BMR), from the background response level. This is a key difference to the NOAEL, which refers to a level with no observable effect, which may be biased by statistical power, or experimental sensitivity (*Brescia, 2020*).

BMRs of 5 and 10% are often used,; however, appropriate BMRs for different endpoints need to be discussed within the (eco)toxicological community (*Slob, 2017*; *Haber et al., 2018*; *Jensen, Kluxen & Ritz, 2019*; *Kluxen, 2020*), as they are not scientifically reasoned, which is similar to the use of $p < 0.05$ (*Salsburg, 2001*; *Kennedy-Shaffer, 2019*). BMDs are essentially similar to effective doses or concentrations but often located in more data-sparse regions of the dose–response curve. In practice, reference values or limit values are derived from the BMD lower limit (BMDL), which is defined as the lower limit of a one-sided confidence interval of the BMD estimate (see, Fig. 1).

Currently, mainly two comprehensive software packages are available for BMD analysis: BMDS developed by US Environmental Protection Agency (EPA) (*Davis, Gift & Zhao, 2011*) and PROAST developed by the Netherlands National Institute for Public Health and the Environment (RIVM) in collaboration with Open Analytics (*Varewyck & Verbeke, 2017*; *Slob, 2018*). The two packages differ in several ways. BMDS includes several dose–response models, but it (for now) only includes Bayesian model averaging and for binary data only. In contrast, PROAST offers model averaging for both binary and continuous response data but includes fewer dose–response models. The two packages also differ in how BMDL is derived. A few other more specialized packages have also been developed for BMD analysis (*Yang, Allen & Thomas, 2007*; *Wheeler & Bailer, 2008*; *Shao & Shapiro, 2018*).

Over the past 25 years, the open-source environment **R** (*R Core Team, 2018*) has developed into an extremely powerful statistical software where many extension packages
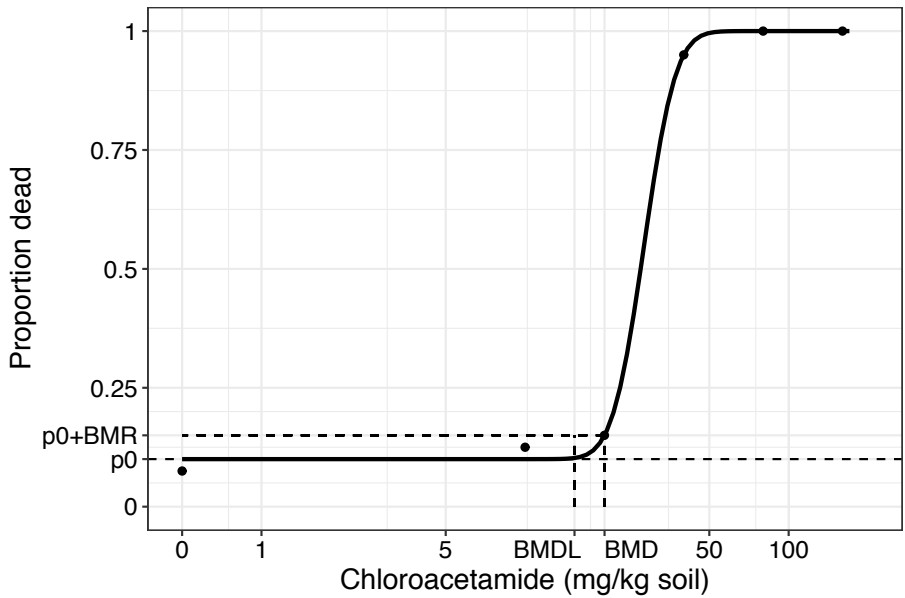

**Figure 1** **The fitted concentration–response curve for the three-parameter log-normal model fitted to data from the earthworm toxicity test, with different concentrations of chloroacetamide (mg/kg soil)** (*Hoekstra, 1987*). The model is shown together with data (mean values per concentration), and the estimated benchmark dose (BMD) and benchmark dose lower limit (BMDL) based on the added risk definition and a benchmark response (BMR) of 0.05. p0 is the estimated probability of dying for the background population, i.e., the background response level.

offer specialized functionality. One such extension package is *drc* developed for the analysis of dose–response data (*Ritz et al., 2019*).

The present paper's objective is to introduce the **R** extension package *bmd* and help (eco)toxicologists with using the tool by the evaluation of multiple case studies and simple example code. The package allows using the most current statistical methods for BMD estimation in **R.** It can be seen as a flexible alternative to BMDS and PROAST in terms of available models, a higher degree of user-control over models, and more approaches for BMDL estimation and model averaging. The option of using sandwich variance–covariance estimates to adjust for model misspecifications is an important addition. The **R** package also offers the possibility to work with count data. An option not available in any other BMD software. The other clear distinction of *bmd* is that it can be used within the **R** environment and thus supplement other statistical analyses within the same software. Further, other extensions that are or will be developed for **R** can be directly used together with *bmd*. This flexibility is currently not achieved in other BMD software.

## BMD ESTIMATION

Estimating a BMD and its corresponding BMDL depends on the type of response data considered, the definition used for defining BMD, the predefined BMR, the choice of dose–response model, and the approach used for estimating BMDL. The dose–response model is fitted to data using *drc*. Since *drc* uses powerful self-starter functions, the fitting

process is automated. The fit can then be visually assessed, and the distribution of the residuals investigated. The fit can also be compared to other candidate models, and model averaging may be performed. Below, the individual steps of the process are described.

## Defining BMD for a predefined BMR

The BMD is defined as the dose resulting in a pre-specified small change, denoted the BMR, in the response relative to the control group's response level. In practice, defining BMD and specifying BMR goes hand in hand (*Jensen, Kluxen & Ritz, 2019*). Different types of changes in the response may be relevant, i.e., absolute versus relative changes compared to the control response level, and accordingly, different definitions of the BMD exist. However, not all definitions are relevant to all types of endpoints. The definitions available in the package *bmd* are described below in detail and are summarized in Table S1. The definitions below are for increasing dose–response curves; for decreasing dose–response curves, the same principles apply, but with a slight change in the definitions (for example, the definition of additional risk for decreasing dose–response curves is: $BMR = p_0 - f(BMD, \beta)$).

### Binomial response data

For binomial response data, two definitions are commonly used (*Jensen, Kluxen & Ritz, 2019*); additional and excess risk (*Piegorsch, 2010*):

Additional risk

$$BMR = f(BMD, \beta) - p_0 \tag{1}$$

Excess risk

$$BMR = \frac{f(BMD, \beta) - p_0}{1 - p_0} \tag{2}$$

$f$ denotes the dose–response model, and $\beta$ is the vector of parameters in the model. $p_0$ represents the mean background response level, i.e., the probability of an adverse effect. The background response level is typically estimated from the model as $p_0 = \lim_{dose \to 0} f(dose, \beta)$ but may also be estimated from a control group or be specified based on prior knowledge (*Jensen, Kluxen & Ritz, 2019*). Notice that if the background response level, $p_0$, equals 0, the two definitions, additional risk (Eq. (1)), and excess risk (Eq. (2)), are identical. Excess risk is the most common definition used for binomial response data. The excess risk definition (Eq. (2)) is recommended by both the US EPA and the European Food Safety Authority (EFSA) (*US Environmental Protection Agency, 2012*; *Hardy et al., 2017*). Appropriate BMR values should ideally be based on expert biological or toxicological knowledge about the test system, but in practice, the decision is often driven by statistical considerations (*Jensen, Kluxen & Ritz, 2019*). US EPA and EFSA recommend choosing a BMR in the lower end of the specific data set's observable dose range. A BMR of 10% using the excess risk definition is often within the dose-range and a common choice for reporting results of a BMD analysis with binomial data (*US Environmental Protection Agency, 2012*; *Hardy et al., 2017*; *Jensen, Kluxen & Ritz, 2019*).

### Continuous response data

For continuous response data, the analogous definitions of additional and excess risk are based on the so-called hybrid approach (*Gaylor & Slikker, 1990*; *Budtz-Jørgensen, Keiding*

*& Grandjean, 2001*; *Jensen, Kluxen & Ritz, 2019*). As continuous data typically reflects a response and not an adverse effect, the hybrid approach "transform" the continuous response measurements to the probability of observing an adverse effect by using a cut-off. The hybrid approach is defined as in e.g., *Jensen, Kluxen & Ritz, 2019*. Specifically, the hybrid approach defines BMD as the dose corresponding to a predefined increase (BMR) in the probability of the response values falling below (or exceeding) a certain cut-off value on the response scale, i.e., the cut-off divides the continuous response scale into normal and abnormal values. In contrast to binomial data, the background response level, $p_0$, of abnormal responses is not directly given by the data but has to be specified using a pre-specified cut-off, possibly based on historical data (*Jensen, Kluxen & Ritz, 2019*). Assuming a normal distribution for the unexposed (background) population the background probability of an adverse event, $p_0$, may be specified as

$$p_0 = 1 - \Phi\left(\frac{x_0 - f(0, \beta)}{\sigma}\right), \tag{3}$$

where $x_0$ is the organism response level considered to be adverse, $\Phi$ denotes the cumulative distribution function for a standard normal distribution, and $\sigma$ is the standard deviation for the control group. With this definition of the background response level (Eq. (3)), the hybrid approach leads to the definition of the BMD as the dose solving the following equations:

Hybrid approach (additional definition)

$$BMR = 1 - \Phi\left(\frac{x_0 - f(BMD, \beta)}{\sigma}\right) - p_0 \tag{4}$$

Hybrid approach (excess definition)

$$BMR = \frac{1 - \Phi\left(\frac{x_0 - f(BMD, \beta)}{\sigma}\right) - p_0}{1 - p_0} \tag{5}$$

For some outcomes, it may be a "natural" choice to choose a specific level, $x_0$, on the response scale to define the cut-off for adverse response levels. However, the most common case is to use a cut-off in terms of a pre-specified number of standard deviations, circumventing the need to decide on an absolute level. For example, US EPA recommends using a cut-off of 1 standard deviation (*US Environmental Protection Agency, 2012*). Both options for defining the background level are available in *bmd*. When using the hybrid approach (Eq. (4)), typical values of BMR are 5% or 10% (*US Environmental Protection Agency, 2012*).

### Count or continuous response data
Additional definitions that are also used in the literature were included in *bmd* available for both continuous and count response data (*Wheeler & Bailer, 2009*; *Davis, Gift & Zhao, 2011*; *Slob, 2017*):

Added response

$$BMR = f(BMD, \beta) - f(0, \beta) \tag{6}$$

Extra response

$$\text{BMR} = \frac{f(\text{BMD}, \beta) - f(0, \beta)}{f(\infty, \beta) - f(0, \beta)} \tag{7}$$

Relative response

$$\text{BMR} = \frac{f(\text{BMD}, \beta) - f(0, \beta)}{f(0, \beta)} \tag{8}$$

In definitions (6)–(8), BMR is defined as an absolute or relative change in the response from the background response level $f(0, \beta)$, as estimated by the model. The added response definition expression mathematically corresponds to the additional risk definition for binomial data where the response is an adverse event. Suppose the upper limit is 1 (for increasing dose–response models or the lower limit is 0 for decreasing dose–response relationships). In that case, the extra response definition corresponds mathematically to the excess risk definition for binomial data. The relative response definition (Eq.(8)) is sometimes referred to as deriving the critical effect size (*Slob, 2017*). US EPA recommends reporting the critical effect size and results from the hybrid approach using BMR values of 5% or 10% as appropriate (*US Environmental Protection Agency, 2012*). EFSA recommends reporting the critical effect size with the default BMR value of 5% (*Hardy et al., 2017*).

### *All types of response data*

Finally, it is an option in *bmd* to specify directly the response level for which to find the dose.

Directly defined

$$\text{BMR} = f(\text{BMD}, \beta).$$

## Estimation

All types of BMD estimates are calculated in an after-fitting step when using the package *bmd*. After-fitting refers to deriving estimates and corresponding standard errors for parameters that do not directly enter the model parameterization and hence have to be estimated in a subsequent step using the delta method. The BMD estimating function in *bmd* is called bmd() and takes a *drc* object (model fit) as its first argument. The alternative to after-fitting would be to re-parameterize the model to include the BMD of interest as a model parameter. The after-fitting approach has, however, the advantage that it suffices to fit the dose–response model once in a parameterization that has proven to be the most robust for estimation (*Ritz et al., 2015*). Accordingly, after-fitting provides an improvement over the sometimes quite unstable re-parameterization approach. This is especially the case when estimating a BMD corresponding to a very low BMR value, which may then be in an area with little or no information in data.

## Choice of dose–response model

A key distinguishing feature of *bmd* is that it inherits flexibility in available models from the package *drc* (*Ritz et al., 2019*). Among others, the parametric dose–response models available include the well-known and often used multi-stage, log–logistic and Weibull models and the flexible class of fractional log–logistic models proposed by *Namata et*

*al. (2008)*. As a non-parametric alternative, the function bmdiso() monotonizes the sequence of response values based on the pool-adjacent-violators algorithm. Based on the monotonized sequence, linear interpolating splines are used to build an isotonic regression model as the dose–response model used for BMD estimation (*Piegorsch et al., 2012*; *Piegorsch et al., 2014*; *Lin, Piegorsch & Bhattacharya, 2015*). Table S2 lists all models from *drc* available for BMD estimation in *bmd*. All the models shown are available for binomial, continuous, and count response data. Even more model expressions may be estimated by fixing one or more parameters. We refer to *Ritz et al. (2019)* for more details.

## Defining BMDL

Different approaches for estimating the BMDL have been proposed. They can be summarized as three general types; asymptotic approaches, inverse regression and based on bootstrapping. Among the asymptotic approaches, the Wald-type confidence intervals are the most simple. The one-sided Wald-type interval needed to estimate the BMDL is found by combining information on the parameters included in the model by using the delta method. The Wald-type intervals may result in unrealistic negative BMDL estimates, which in practice will be truncated at 0 because of the symmetric confidence interval. Especially in the low dose area, this may be a problematic assumption. One way to work around this is to use a transformation (typically the logarithm) to avoid negative values. Inverse regression reports the BMDL as the dose associated with the upper limit of the confidence band for the change in response reaching the predefined BMR (*Buckley, Piegorsch & West, 2009*; *Fang, Piegorsch & Barnes, 2015*).

Bootstrap methods for estimating BMDL may rely on non-parametric, parametric, and semi-parametric strategies depending on the type of response data. The non-parametric bootstrap is based on resampling with replacement from each dose group of the original data set. The parametric bootstrap for continuous data samples from a normal distribution with dose-specific mean but equal standard deviation, assuming equal variance between groups, i.e., response values are sampled from:

$$\overline{Y_{ij}} \approx N(E(Y_i), SD(Y_0))$$

for observation $j$ in dose-group $i$. In case of binomial data, each bootstrap data set is sampled from a binomial distribution with dose-specific numbers of observations and probability of an event. That is, response values are sampled from:

$$\overline{Y_{ij}} \approx \text{Binom}(N_i, \frac{Y_i}{N_i})$$

where $\frac{Y_i}{N_i}$ denotes the observed proportion of the population in dose group $i$ experiencing the event.

In case a dose has only non-events or only events, shrinkage is used to avoid that the resampling always produces 0 or 1, respectively. In this case, data is sampled from

$$\overline{Y_{ij}} \approx \text{Binom}(N_i, \frac{Y_i + 1/4}{N_i + 1/2})$$

as suggested elsewhere (*Piegorsch et al., 2012*).

Semi-parametric bootstrapping, only available for continuous response data, is done by sampling with replacement from the residuals assuming exchangeability over all observations.

## Exploring model assumptions

For continuous data, residual plots showing standardized residuals against predicted values can be used to assess variance homogeneity. A random scatter plot will support the model assumptions whereas clear patterns in the plot indicate deviations from model assumptions. Assessment of the normality assumption can be based on quantile–quantile plots on the residuals. For an example on how to do a visual assessment of model assumptions, we refer to *Ritz et al. (2019)*. Using the visual assessment of residual and quantile–quantile plots is associated with some degree of subjective evaluation. However, the test-based assessment also has limitations, as it might result in overfitting, or because the assumptions underlying the tests themselves are not fulfilled (*Wang & Riffel, 2011*).

Assessment of over-dispersion for binomial or count data can be done by comparing residual deviance to degrees of freedom. If the residual deviance is higher than the degrees of freedom, there is excess variation in data (over-dispersion) not accounted for by the model. Alternatively, a model allowing an extra variance parameter, i.e., a negative binomial model can be fitted and the need for the extra parameter evaluated by comparing models with and without the extra parameter, for example, using AIC (Akaike information criteria). Finally, the model may be estimated using sandwich estimates (see 'Dealing with distributional misspecification'). A large change in the standard errors indicate over-dispersion in data.

## Dealing with distributional misspecification

Extreme or, by other means, deviating observations may be handled already in the model-fitting step using robust estimation in *drc*. The resulting variance–covariance matrix will be propagated to the subsequent after-fitting step where BMD and BMDL are derived.

When assumptions regarding normality and/or the variance homogeneity are not satisfied, consistent estimates of the standard errors can be obtained by adjusting the estimated variance–covariance matrix. One way of doing this is to use a modified variance–covariance matrix defined as:

$$\text{var}\left(\hat{\beta}\right) = \hat{A}^{-1}\hat{B}(\hat{A}^{-1})^T$$

where $\hat{A}$ is the usual estimate of the variance based on the information matrix, and $\hat{B}$ is a correction term based on the first derivatives on the log-likelihood function. Due to the form of the modified variance–covariance matrix, the resulting adjusted standard errors are referred to as sandwich estimates. Notice that the sandwich approach only modifies standard errors but leaves estimated model parameters unaltered. Accordingly, the underlying assumption of a correctly specified mean function remains.

Finally, log-normal distributed data may be handled by log-transforming the response variable before fitting the dose–response model, which leads to a change in the interpretation of BMD and BMDL, now relating to a pre-determined change in the response on log-scale. Alternatively, the transform-both-sides approach could be used

such that the assumed dose–response relationship, and accordingly, BMD and BMDL will retain their original interpretation (*Carroll & Ruppert, 1988*).

## Model averaging

Interpolation in a dose region with little or no data, as is typically the case for BMD analysis, is highly dependent on the fit and choice of dose–response model. To partly overcome this issue, it has been proposed to evaluate several models and subsequently select the best model determined by means of some goodness-of-fit criterion (*Slob, 2002*). However, the uncertainty pertaining to the model selection process is not incorporated in the BMD and the associated BMDL. Indeed, results based on a best fitting model found using a model selection procedure may result in biased estimates of BMD and non-protecting estimates of BMDL, i.e., too high BMDL estimates with coverage below the nominal level (*West et al., 2012*; *Ringblom, Johanson & Öberg, 2014*). As a consequence, model averaging has been repeatedly advocated for BMD analysis by several authors (e.g., *Faes et al., 2007*; *Jensen & Ritz, 2015*; *Kang, Kodell & Chen, 2000*; *Namata et al., 2008*; *Wheeler & Bailer, 2007*; *Wheeler & Bailer, 2008*) as well as regulating authorities (*Hardy et al., 2017*). The use of model averaging has the consequence that the flexibility of the assumed candidate models to some extent replaces lack of data. The package *bmd* offers a large set of dose–response models and accordingly a high flexibility of the candidate set of models for model averaging. Another advantage is that it allows for an automated approach once the candidate models are specified.

Two different approaches may be used for estimating BMD by model averaging. One approach is to make a weighted average of the BMD estimates from all the candidate models (e.g., *Bailer, Noble & Wheeler, 2005*; *Kang, Kodell & Chen, 2000*; *Moon et al., 2005*; *Namata et al., 2008*). The other is to average entire curves and then find the BMD as the dose associated with the change of interest in the response generated by this weighted averaged model, i.e., using the appropriate definition (*Wheeler & Bailer, 2007*; *Wheeler & Bailer, 2008*). Weights are usually based on some measure of goodness of fit. A common choice is the AIC weights defined as:

$$w_k = \frac{\exp\left(\frac{-\Delta_k}{2}\right)}{\sum_{i=1}^{K} \exp\left(\frac{-\Delta_i}{2}\right)}. \tag{9}$$

Here $K$ denotes the total number of candidate models and $\Delta_k = \text{AIC}_k - min_i \text{AIC}_i$ is the AIC-difference for model $k$. Alternatively, weights based on the Bayesian information criteria (BIC) (*Claeskens & Hjort, 2008*) or user-defined weights may be provided.

Estimating the BMDL corresponding to a model-averaged BMD is less straightforward and has accordingly been the subject of much methodological research (*Jensen, Kluxen & Ritz, 2019*). Three different approaches are available in *bmd*.

The first option is using a one-sided Wald confidence intervals with standard error approximated by a variance inequality by *Buckland, Burnham & Augustin (1997)*:

$$\text{var}(\text{BMD}_{MA}) \leq \left(\sum_{k=1}^{K} w_k \cdot \sqrt{\text{var}(\text{BMD}_k) + (\text{BMD}_k - \text{BMD}_{MA})^2}\right)^2.$$

Here, $BMD_k$ is the BMD for model k and $BMD_{MA}$ is the weighted average of BMD estimates. The second option is using a weighted average of the BMDLs from all the candidate models, in the same way, BMD estimates are averaged to get the model-averaged BMD (*Kang, Kodell & Chen, 2000*):

$$\text{BMDL}_{MA} = \sum_{k=1}^{K} w_k \cdot \text{BMDL}_k$$

with weights, $w_k$, specified as above, e.g., using Eq.(9). Finally, different bootstrap approaches may be used to estimate BMDL corresponding to the model-averaged BMD. For model averaging based on entire curves, the only option for estimating the BMDL is to use bootstrap.

### Hierarchical designs

Data obtained from designs with a hierarchical or nested structure do not comply with the basic assumptions of independence underlying the classical dose–response regression analysis. Hence, sticking to these models may result in biased BMDL estimates, i.e., too high or low BMDL estimates with coverage far from the nominal level, usually 95%. Preferably, a mixed effects dose–response model should be used to model such data in order to take into account the correlation structure in data (*Ritz, Gerhard & Hothorn, 2013*). An alternative approach could be to adjust the estimated standard errors of BMD using sandwich estimates. Another alternative is to fit dose–response models for each independent subsample, and combine the estimates using a meta-analytic approach (*Ritz et al., 2019*). This latter approach has been shown to result in estimates with stable and near nominal coverage for the closely related effective doses (*Jiang & Kopp-Schneider, 2014*), and it can therefore be used as an attractive alternative to the more complex fitting of mixed effects dose–response models. An example of this meta-analytic approach is provided in the last of the five examples given below. A similar two-step meta-analytic approach could be used to integrate historical data (*Ritz et al., 2019*; *Jensen, Kluxen & Ritz, 2019*).

## EXAMPLES

The **R** package *bmd* builds on the flexibility of *drc*, and thereby facilitates BMD and BMDL estimation for a wide range of dose–response models. It was first mentioned as a function in *Ritz et al. (2015)* but has been substantially updated since 2015 with the one common denominator being that both versions utilizes the functionality available in *drc*. The main function in the package *bmd* is bmd(), which uses a *drc* object (a dose–response model fit) and can be easily extended by including arguments. In the following, we revisit five data examples from the literature. **R** code for the different examples is provided as supplementary material. In short, after installation of the **R** package from GitHub, the basic approach is to fit a curve using drm(),

```
library(drc)
modelfit <- drm(...)
```

and then apply the function bmd()

```
library(bmd)
```

```
bmd(modelfit, bmr = ..., backgType = ''...'', def = ''...'')
```
with a specification of BMR, the background type (usually "modelbased"), and the type of definition as indicated in Table S1.

### Binomial data: an earthworm toxicity test with chloroacetamide

Our first example revisits data from an earthworm toxicity test (*Hoekstra, 1987*). For each of six concentrations (including a zero control), 40 earthworms were exposed to the herbicide chloroacetamide, and the resulting number of dead earthworms was counted. The control or natural mortality was 3/40. Following the authors, we fitted a three-parameter log-normal model to the data, estimating the natural mortality as a lower limit. Figure 1 shows the resulting model fit. The model-based estimated natural mortality was 0.10 (95% CI [0.03–0.17]).

In this example, we considered a BMR of 5%, as this small change in probability of an adverse event (death) was well covered by the dose-range. The BMD corresponding to a BMR of 5% using the additional risk definition was the dose associated with a proportion of dead earthworms equal to

$$BMR + p_0 = 0.10 + 0.05 = 0.15.$$

The resulting estimated BMD and BMDL were 20.02 mg/kg and 15.40 mg/kg, respectively. The BMD corresponding to a BMR of 5% using the excess risk definition was the dose associated with a proportion of dead earthworms equal to

$$BMR \cdot (1 - p_0) + p_0 = 0.05 \cdot (1 - 0.10) + 0.10 = 0.145.$$

The resulting estimated BMD and BMDL for the excess risk definition were 19.79 mg/kg and 15.15 mg/kg, respectively.

### Count data: a toxicity test in aquatic plants

As part of an experiment examining the toxicity of metsulfuron-methyl on different aquatic plant species, *Cedergreen, Streibig & Spliid (2004)* considered the effect on *Elodea canadensis*. Vegetative shoots were placed in an aquarium growth cabinet with a photoperiod of 16 h and at day/night water temperatures of 18/15 °C. Six *E. canadensis* shoots were exposed to each of seven different concentrations of metsulfon-methyl (0, 0.01, 0.1, 1, 10, 100, 1,000 µg/L medium) in an artificial nutrient medium. Plants were harvested after 14 days and lateral shoots counted among other endpoints.

A three-parameter log–logistic model was fitted to the shoot counts assuming Poisson distributed data. Figure S1 shows the resulting model fit. The figure also shows all data points making it clear that data contains several 0s and also other ties. The fitted model described data adequately, as also indicated by a residual plot (shown in the **R** code in the Supplemental Information).

We considered an absolute change in the response of one less shoot for these data, as biological implications on 1 less shoot can be considered relevant. That is, BMD was estimated with the added definition and a BMR of 1 shoot. Using nonparametric bootstrap, we found BMD equal to 0.133 µg/L and BMDL equal to 0.052 µg/L.

If we ignored that these data were count data and instead carried out the analysis assuming a normal distribution, the resulting BMD and BMDL would be 0.254 µg/L and 0.092 µg/L, respectively. Ignoring that these were count data would result in less protective estimates in terms of a much higher BMD and a somewhat higher BMDL that was based on much wider confidence intervals.

## Count data: a toxicity test with copper under varying temperature

*Cedergreen et al. (2016)* examined the effect of varying temperature on the toxicity of copper on the nematode *Caenorhabditis elegans* (Maupas). As part of the experiment, 12 nematode worms were exposed to each of five copper concentrations (1, 3, 8, 20, 40 mg/L agar) under varying temperatures with daily fluctuations of ±4 degrees around a mean temperature of 20 °C. Besides, 36 worms were exposed to the same temperature fluctuations but were considered controls with no copper exposure. During the experiment's reproduction phase, the nematodes were moved to new wells in fresh plates every day. It allowed registration of the daily egg production and the lifespan of each nematode. The number of fertile eggs and hatched juveniles were counted as offspring.

We followed the guidelines for continuous data from US EPA and estimated the BMD associated with a BMR of 10% using the relative definition, i.e., the concentration associated with a 10% reduction in the total number of offspring relative to the control group. A three-parameter log–logistic model was fitted to data under the assumption of Poisson distributed data. Figure S2 shows the resulting model fit. The resulting BMD and BMDL were 10.47 mg/L agar and 9.40 mg/L agar, respectively.

It may be argued that the above model is problematic due to premature mortality (*Delignette-Muller et al., 2014*). An alternative would accordingly be to use a weighted dose–response model taking into account the lifespan of the nematodes. The BMD from this new model should now be interpreted as the concentration associated with a 10% change in the number of offspring per day (the lifespan unit) relative to the control group. Because of the two different interpretations, the BMD and BMDL estimates from this model (31.16 mg/L and 28.89 mg/L, respectively) cannot be directly compared to those from the first model.

A second issue related to reproduction data is that count data often exhibit overdispersion, i.e., more variation than can be explained by the model. Two different approaches can be used to overcome this challenge; one is to change the underlying assumed distribution, the other to adjust the standard errors. The first option would, for example, be to use a negative binomial distribution instead of the common choice of the Poisson distribution. The resulting BMD and BMDL were then 27.99 mg/L and 15.99 mg/L, respectively. The second option is to use sandwich estimates when estimating the BMDL while trusting that the dose–response model function appropriately describes the mean trend in data. The result of this approach was a BMD of 31.16 mg/L and a BMDL of 18.63 mg/L. Notice that both procedures accounting for overdispersion resulted in a large decrease of the BMDL, indicating that overdispersion was indeed present in these data.

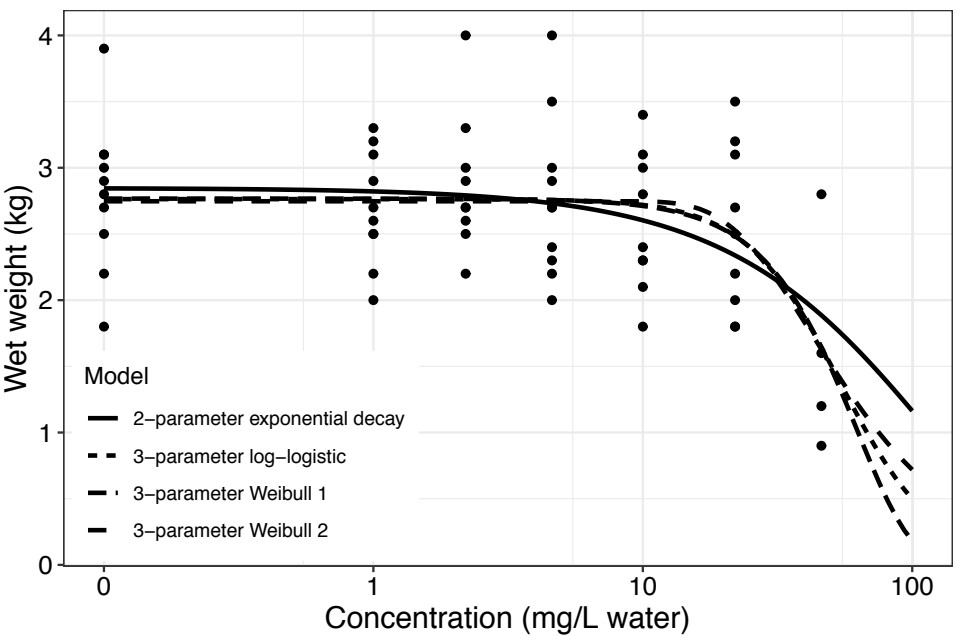

**Figure 2** Fitted concentration–response curve for the two-parameter exponential decay model, the three-parameter log–logistic model and the two different Weibull models fitted to the Rainbow trout data, (*Organisation for Economic Co-operation and Development(OECD), 2006*). The curve is shown together with all data points. The Rainbow trouts were exposed to different concentrations (mg/L water) of an unknown agent.

## Continuous data: a fish test

OECD describe data from a 21 day fish test following the guidelines by OECD GL204 and using the test organism Rainbow trout *Oncorhynchus mykiss* (*Organisation for Economic Co-operation and Development(OECD), 2006*). The Rainbow trouts were held in 14−15 °C water and exposed to one of 7 concentrations (6 nonzero concentrations + control) of an unknown agent. After 28 days, the wet weight was registered. There were ten replicates per concentration. However, for some higher doses, missing values occurred, resulting in a final data set with only 61 observations of the weight. Ignoring such missing values may lead to biased results, depending on the mechanism causing the missing values, but we will ignore the potential problem here for illustrating the BMD methodology.

Following *OECD (2006)*, we fitted a two-parameter exponential decay model to the Rainbow trout data. Figure 2 (full line) reveals an appropriate fit to data supported by residual and QQ plots (see supplementary material: **R** code for the examples).

For a BMR of 5% using the hybrid approach with 2 standard deviations as the cut-off the BMD and BMDL were 4.23 mg/L and 2.13 mg/L, respectively.

While the two-parameter exponential decay model showed a good fit to data, other models may give a similar fit to data. The three-parameter log–logistic and two different three-parameter Weibull models showed similar reasonable visual fit to data (see Fig. 2). Comparing the models using AIC also showed a similar fit of the four models (Table 1).

**Table 1   Resulting benchmark dose (BMD), benchmark dose lower limit (BMDL), Akaike information criteria (AIC), and weight based on AIC for four different models fitted to data from a fish test with Rainbow trout.** BMD was estimated based on the hybrid approach using 2 standard deviations as the cut-off and BMR = 0.5. The model-averaged BMDLs were based on non-parametric bootstrap.

|  | Parameters | BMD (mg/L) | BMDL (mg/L) | AIC | Weight |
|---|---|---|---|---|---|
| Exponential decay | 2 | 12.65 | 6.37 | 106.31 | 0.333 |
| Log–logistic | 3 | 22.91 | 9.05 | 106.65 | 0.237 |
| Weibull type 1 | 3 | 22.72 | 8.12 | 106.58 | 0.253 |
| Weibull type 2 | 3 | 24.26 | 14.26 | 106.94 | 0.177 |
| Model averaging (estimate averaging) |  | 19.68 | 8.04 |  |  |
| Model averaging (curve averaging) |  | 20.35 | 7.73 |  |  |

Model averaging based on the four models using non-parametric bootstrap for estimating BMDL resulted in a BMD much higher and a BMDL rather close to the corresponding estimates from the original (exponential decay) model. The alternative model averaging approach where entire curves were averaged before finding BMD and BMDL resulted in a slightly higher BMD but a slightly lower BMDL.

The computation time (using a standard Dell laptop) for 1000 bootstrap samples using the four models was app. 3 min for the estimate averaging and app. 5 min for the curve averaging. For comparison, the computation time for this data set using PROAST (web application) was app. 6 min for model averaging based on four models and 1000 bootstrap samples.

## Binomial data in a hierarchical design: an acute toxicity test with α-cypermethrin

*Gottardi & Cedergreen (2019)* investigated the toxicity of α-cypermethrin on *Daphnia magna* using an acute toxicity test following OECD guidelines for testing chemicals (*Organisation for Economic Co-operation and Development (OECD), 2012b*). In nine independent experiments conducted at different times, *D. magma* neonates (<1 day old) were exposed to different concentrations of α-cypermethrin or acetone control for 48 h in M7 medium. Four replicates of five organisms were used for each concentration. The number of concentrations used in each sub-experiment varied from 6 to 7 (5 or 6 and control).

For this example, the purpose was to estimate BMD and BMDL for a BMR of 5% using the excess risk definition as recommended by EFSA and US EPA. However, due to the hierarchical design with sub-experiments, a simple dose–response model may not appropriately capture the dependence structure in data. Consequently, a two-step meta-analytic approach was applied. In the first step, a two-parameter log–logistic model was fitted to data from each sub-experiment. From each model fit, the estimated BMD and the corresponding standard error were extracted. In the second step, a random effects meta-analytic model was fitted using the **R** package *metafor* (*Viechtbauer, 2010*).

 

Figure S3 shows the resulting model fits from step one. The estimated BMD and BMDL resulting from the meta-analytic model in step 2 were 0.07 µg/L and 0.05 µg/L, respectively.

## VALIDATION OF THE R PACKAGE

A small-scale validation of the **R** package was carried out by analyzing a number of datasets using *bmd* and BMDS version 3.2.0 from the US. EPA and the PROAST web application (https://shiny-efsa.openanalytics.eu/app/bmd) from EFSA.

### Methods

For binomial data, the three software packages were compared for three data sets, four different models, and two BMR levels. The three data sets used were data from experiment 4, 7, and 9 from the last of the five examples corresponding to different dose–response shapes (see Fig. S4). These data were analyzed using a log–logistic, a log-normal, and a Weibull model. For BMDS and *bmd*, two-parameter models were used, assuming lower and upper limits to be 0 and 1, respectively. In PROAST, it was not possible to fit models with a lower limit of 0, instead three-parameter models were fitted. BMR was set to either 5% or 10%, and BMD was defined using the excess risk definition. BMDL was estimated using profile likelihood for both BMDS and PROAST, while three different alternatives were used for *bmd:* delta method-based Wald confidence intervals, confidence intervals obtained from inverse regression, and non-parametric bootstrap confidence intervals obtained using resampling within dose groups from the original data set. BMDL was found as the 5% percentile in the bootstrap distribution.

For continuous data, the three software packages were compared for nine data sets, two models, and two BMR levels. The nine data sets were simulated from three four-parameter log–logistic models with similar lower and upper limits (2 and 10, respectively) but different location and shape parameters. The three different models are shown in Fig. S4. For each model, data were generated at five doses (0.1, 0.5, 1, 5, and 10), assuming normal distributions with (i) 10 replicates and a standard deviation of 1, (ii) 10 replicates and a standard deviation of 0.1, or (iii) 3 replicates and a standard deviation of 0.1. These data were analyzed using a four-parameter log–logistic model (called the Hill model in BMDS and PROAST) and a four-parameter Weibull model (called the exponential model in BMDS and PROAST). BMR was set to either 0.05 or 0.1, and BMD was defined using the relative risk definition. BMDL was estimated using profile likelihood for both BMDS and PROAST. For *bmd*, three alternatives for estimating BMDL were applied: delta-method-based Wald confidence intervals, confidence intervals obtained from inverse regression, and semi-parametric bootstrap confidence intervals obtained through resampling of residuals from the dose–response model assuming exchangeability over all observations. BMDL was found as the 5% percentile in the bootstrap distribution.

We also compared *bmd* to BMDS and PROAST on five data sets used in the EFSA report on benchmark dose modeling (*Hardy et al., 2017*). These data sets are described in detail below.

Data set 1 contains continuous data from a subchronic National Toxicity Study 416 for an unknown agent. Specifically, bodyweight mean values and standard deviations are

available for six dose groups of 10 animals (mg/kg body weight), including a control group. Data were analyzed using a three-parameter exponential model, the relative definition, and a BMR of 5%.

Data set 2 contains binomial data from a toxicity study examining the incidence of gastric impaction from an unknown agent. Ten animals were used in each of four dose groups (mg/kg body weight), which included a control group. Data were analyzed using a three-parameter log–logistic model (with an estimated lower limit), the excess risk definition, and a BMR of 10%.

Data set 3 comes from human dose–response study, where each individual has its exposure limit. The binary endpoint is normal or abnormal eye-hand coordination in individual workers exposed to different levels of CRD (unit unknown). Data were analyzed using a one-stage model, the excess risk definition, and a BMR of 10%.

Data set 4 comes from a two-year study in male mice where body weight is reported as mean values, standard deviations, and sample sizes for each of four dose groups (mg/kg body weight) of an unknown agent, including a control. Data were analyzed using a three-parameter Hill model and a three-parameter exponential model using the relative definition and a BMR of 5%.

Finally, data set 5 contains binomial data from a two-year study in female rats examining thyroid epithelial vacuolization incidence. Four dose groups (mg/kg body weight) of an unknown agent, including a control, were considered. Data were analyzed using a three-parameter log–logistic and a three-parameter log-normal model, using the excess risk definition, and a BMR of 10%.

## Results

The results for binomial and continuous data are shown in Tables 2 and 3, respectively, while the results for the five data sets used in the EFSA report are presented in Table 4. For the binomial data, PROAST and *bmd* agreed on BMD estimates to the third significant number for all data examples and models. BMDS agreed with *bmd* for most of the models, except the Weibull model in a few cases. For continuous data, *bmd* and BMDS agreed on the BMD estimate to the third significant number with only a few exceptions. PROAST gave quite different results for most scenarios.

BMDL estimates from *bmd* did not match those from BMDS and PROAST as these were based on different methods. However, for binomial data the delta method derived BMDL estimates from *bmd* were generally close to the BMDL estimates from both PROAST and BMDS. For continuous data, the bootstrap BMDL estimates from *bmd* were generally very close to BMDL estimated by BMDS. From the results presented here, it is not possible to say which of the different approaches performs best.

The running time for some of the functionality involving bootstrap may be rather long. However, it is mainly an issue for simulation purposes where hundreds or thousands of estimates are to be found. In addition, comparisons to PROAST web-application shows shorter run times for *bmd*.

**Table 2  Estimated benchmark dose (BMD) and benchmark dose lower limit (BMDL) for different binomial data sets, different models and different levels of BMR using PROAST, BMDS and *bmd*.** PROAST and BMDS uses profile likelihood intervals for estimating BMDL while the R package *bmd* uses the delta method, inverse regression or bootstrap. For all data sets the excess risk definition was used.

| Data set | Model[a] | BMR | BMD | | | BMDL | | | | |
|---|---|---|---|---|---|---|---|---|---|---|
| | | | PROAST | BMDS | *bmd* | PROAST profile | BMDS profile | *bmd* delta | *bmd* inverse | *bmd* bootstrap |
| A | Log–logistic | 0.05 | 0.044 | 0.044 | 0.044 | 0.022 | 0.022 | 0.020 | 0.029 | 0.027 |
| | | 0.1 | 0.063 | 0.062 | 0.062 | 0.036 | 0.036 | 0.036 | 0.043 | 0.042 |
| | Log-normal | 0.05 | 0.047 | 0.047 | 0.047 | 0.026 | 0.026 | 0.025 | 0.033 | 0.030 |
| | | 0.1 | 0.063 | 0.063 | 0.063 | 0.038 | 0.038 | 0.038 | 0.045 | 0.043 |
| | Weibull | 0.05 | 0.027 | 0.027 | 0.027 | 0.010 | 0.010 | 0.006 | 0.014 | 0.013 |
| | | 0.1 | 0.047 | 0.047 | 0.047 | 0.022 | 0.022 | 0.018 | 0.027 | 0.026 |
| B | Log–logistic | 0.05 | 0.087 | 0.087 | 0.087 | 0.063 | 0.063 | 0.064 | 0.071 | 0.072 |
| | | 0.1 | 0.102 | 0.102 | 0.102 | 0.079 | 0.079 | 0.080 | 0.085 | 0.086 |
| | Log-normal | 0.05 | 0.089 | 0.089 | 0.089 | 0.066 | 0.066 | 0.066 | 0.074 | 0.075 |
| | | 0.1 | 0.102 | 0.102 | 0.102 | 0.079 | 0.079 | 0.080 | 0.085 | 0.087 |
| | Weibull | 0.05 | 0.078 | 0.072 | 0.078 | 0.049 | 0.049 | 0.049 | 0.058 | 0.059 |
| | | 0.1 | 0.098 | 0.093 | 0.098 | 0.069 | 0.068 | 0.069 | 0.076 | 0.078 |
| C | Log–logistic | 0.05 | 0.055 | 0.55 | 0.055 | 0.032 | 0.031 | 0.029 | 0.037 | 0.035 |
| | | 0.1 | 0.081 | 0.081 | 0.081 | 0.052 | 0.052 | 0.050 | 0.058 | 0.056 |
| | Log-normal | 0.05 | 0.058 | 0.058 | 0.058 | 0.036 | 0.036 | 0.034 | 0.041 | 0.039 |
| | | 0.1 | 0.080 | 0.080 | 0.080 | 0.053 | 0.053 | 0.052 | 0.058 | 0.057 |
| | Weibull | 0.05 | 0.046 | 0.046 | 0.046 | 0.024 | 0.024 | 0.020 | 0.029 | 0.028 |
| | | 0.1 | 0.077 | 0.077 | 0.077 | 0.046 | 0.046 | 0.043 | 0.051 | 0.051 |

**Notes.**
[a] All models fitted as unrestricted models in BMDS.

## DISCUSSION

We have described in detail the functionality of the **R** package *bmd* for BMD estimation. The usefulness and flexibility of *bmd* were illustrated by means of a number of examples, covering both simple and more complex data structures. In the first example, binomial data for a single dose–response curve was used to estimate BMD and BMDL. In the second and third example, we revisited a data set with count data and considered different ways of approaching the related challenges, including potential overdispersion of the counts. The fourth data example illustrated the flexibility of the package to estimate a model-averaged BMD and BMDL, whereas the last example showed how to handle data from more complex experimental designs.

The package *bmd* differs from the existing specialized BMD software packages BMDS and PROAST as it provides BMD estimation for several types of response data, including count data and time-to-event data, two types of data that often occur in ecotoxicology. Count data are often handled by assuming a normal or log-normal distribution ignoring that such data cannot be negative, may present ties and usually experience variances that are a function of the mean and accordingly violates the assumption of variance homogeneity, i.e., for the special case of Poisson distributed variables, the variance equals the mean. Though transformations may accommodate some of the issues related to count data, it has

**Table 3 Estimated benchmark dose (BMD) and benchmark dose lower limit (BMDL) for different continuous data sets, different models and different levels of BMR using PROAST, BMDS and *bmd*.** PROAST and BMDS uses profile likelihood intervals for estimating BMDL while the R package *bmd* uses the delta method, inverse regression or bootstrap. For all data sets, the relative definition of BMD was used.

| Data set | Std | Rep | Model | BMR | BMD | | | BMDL | | | | |
|---|---|---|---|---|---|---|---|---|---|---|---|---|
| | | | | | PROAST | BMDS | *bmd* | PROAST profile | BMDS profile | *bmd* delta | *bmd* inverse | *bmd* bootstrap |
| A | 1 | 10 | Log–logistic[a] | 0.05 | 0.088 | 0.144 | 0.144 | 0 | 0.050 | 0.037 | 0.082 | 0.051 |
| | | | | 0.1 | 0.129 | 0.200 | 0.200 | 0 | 0.080 | 0.077 | 0.119 | 0.080 |
| | | | Weibull[2] | 0.05 | 0.040 | 0.126 | 0.126 | 0 | 0.018 | 0.004 | 0.063 | 0.017 |
| | | | | 0.1 | 0.070 | 0.183 | 0.183 | 0 | 0.034 | 0.038 | 0.097 | 0.032 |
| | 0.1 | 10 | Log–logistic | 0.05 | 0.101 | 0.107 | 0.107 | 0.091 | 0.105 | 0.097 | 0.097 | 0.096 |
| | | | | 0.1 | 0.148 | 0.154 | 0.154 | 0.135 | 0.151 | 0.142 | 0.140 | 0.140 |
| | | | Weibull | 0.05 | 0.049 | 0.074 | 0.074 | 0.041 | 0.060 | 0.061 | 0.061 | 0.061 |
| | | | | 0.1 | 0.086 | 0.115 | 0.115 | 0.073 | 0.097 | 0.099 | 0.098 | 0.098 |
| | 0.1 | 3 | Log–logistic | 0.05 | 0.092 | 0.097 | 0.097 | 0.080 | 0.078 | 0.075 | 0.077 | 0.079 |
| | | | | 0.1 | 0.138 | 0.140 | 0.140 | 0.121 | 0.1156 | 0.114 | 0.114 | 0.117 |
| | | | Weibull | 0.05 | 0.043 | 0.066 | 0.066 | 0.031 | 0.040 | 0.037 | 0.043 | 0.043 |
| | | | | 0.1 | 0.076 | 0.105 | 0.105 | 0.058 | 0.067 | 0.066 | 0.072 | 0.072 |
| B | 1 | 10 | Log–logistic | 0.05 | 0.003 | 0.092 | 0.092 | 0 | 0.005 | 0 | 0.035 | 0.005 |
| | | | | 0.1 | 0.008 | 0.148 | 0.149 | 0 | 0.011 | 0 | 0.06 | 0.012 |
| | | | Weibull | 0.05 | 0 | 0.045 | 0.045 | 0 | 0.015 | 0 | 0.014 | 0.003 |
| | | | | 0.1 | 0.002 | 0.085 | 0.085 | 0 | 0.030 | 0 | 0.029 | 0.007 |
| | 0.1 | 10 | Log–logistic | 0.05 | 0.033 | 0.032 | 0.032 | 0.019 | 0.023 | 0.024 | 0.023 | 0.023 |
| | | | | 0.1 | 0.07 | 0.067 | 0.067 | 0.043 | 0.050 | 0.054 | 0.051 | 0.05 |
| | | | Weibull | 0.05 | 0.008 | 0.056 | 0.016 | 0.003 | 0.051 | 0.011 | 0.011 | 0.011 |
| | | | | 0.1 | 0.022 | 0.112 | 0.038 | 0.012 | 0.102 | 0.029 | 0.027 | 0.027 |
| | 0.1 | 3 | Log–logistic | 0.05 | 0.029 | 0.036 | 0.036 | 0.016 | 0.028 | 0.024 | 0.023 | 0.023 |
| | | | | 0.1 | 0.064 | 0.074 | 0.074 | 0.038 | 0.051 | 0.055 | 0.051 | 0.051 |
| | | | Weibull | 0.05 | 0.008 | 0.056 | 0.019 | 0.004 | 0.049 | 0.012 | 0.012 | 0.012 |
| | | | | 0.1 | 0.023 | 0.113 | 0.044 | 0.012 | 0.099 | 0.031 | 0.029 | 0.03 |
| C | 1 | 10 | Log–logistic[c] | 0.05 | – | 0.717 | 0.717 | – | 0.047 | 0 | 0.261 | 0.079 |
| | | | | 0.1 | – | 0.971 | 0.971 | – | 0.098102 | 0 | 0.369 | 0.155 |
| | | | Weibull | 0.05 | – | 0.666 | 0.666 | – | 0.051 | 0 | 0.224 | 0.067 |
| | | | | 0.1 | – | 0.930 | 0.930 | – | 0.102 | 0 | 0.329 | 0.134 |
| | 0.1 | 10 | Log–logistic | 0.05 | 0.847 | 0.843 | 0.843 | 0.737 | 0.832 | 0.683 | 0.696 | 0.693 |
| | | | | 0.1 | 1.137 | 1.126 | 1.126 | 1.010 | 1.111 | 0.954 | 0.953 | 0.957 |
| | | | Weibull | 0.05 | 0.827 | 0.833 | 0.833 | 0.705 | 0.677 | 0.651 | 0.679 | 0.677 |
| | | | | 0.1 | 1.155 | 1.132 | 1.132 | 1.010 | 0.950 | 0.929 | 0.948 | 0.952 |
| | 0.1 | 3 | Log–logistic | 0.05 | 1.724 | 3.262 | 2.621 | 1.070 | 3.219 | 0 | 1.501 | 0.821 |
| | | | | 0.1 | 2.054 | 3.491 | 2.905 | 1.380 | 3.459 | 0 | 1.684 | 1.107 |
| | | | Weibull | 0.05 | 2.007 | 3.533 | 2.647 | 1.090 | 3.431 | 0 | 1.513 | 0.802 |
| | | | | 0.1 | 2.382 | 3.753 | 2.956 | 1.450 | 3.329 | 0 | 1.716 | 1.095 |

**Notes.**
[a]Hill model for BMDS and PROAST. The Hill model was fitted as an unrestricted model in BMDS.
[b]Exponential model in BMDS and PROAST. The exponential model was fitted as a restricted model in BMDS.
[c]No model fitted for this data set using PROAST.

Jensen et al. (2020), *PeerJ*, DOI 10.7717/peerj.10557

**Table 4  Estimated benchmark dose (BMD) and benchmark dose lower limit (BMDL) for five data sets used in the Benchmark dose report from EFSA (*Hardy et al., 2017*).** Data were analyzed using PROAST, BMDS and *bmd*. PROAST and BMDS uses profile likelihood intervals for estimating BMDL while the R package *bmd* uses the delta method, inverse regression or bootstrap.

| Data set | Data type | Definition | BMR | Model | BMD | | | BMDL | | | | |
|---|---|---|---|---|---|---|---|---|---|---|---|---|
| | | | | | PROAST | BMDS | *bmd* | PROAST profile | BMDS profile | *bmd* delta | *bmd* inverse | *bmd* bootstrap |
| 1 | Continuous | Relative | 5% | Exponential | 235.1 | 233.7 | 235.5 | 170 | 170.4 | 201.5 | 203.3 | 201.1 |
| 2 | Binomial | Excess | 10% | Log–logistic | 399 | 398.6 | 398.7 | 171 | 171.0 | 204.2 | 291.4 | 61.9 |
| 3 | Binary | Excess | 10% | One-stage | 173 | 172.7 | 172.7[a] | 92.4 | 92.3 | 35.4 | 95.2 | 75.2 |
| 4 | Continuous | Relative | 5% | Exponential | 0.297 | 0.302 | 0.304 | 0.198 | 0.229 | 0.112 | 0.162 | 0.277 |
| | | | | Hill[b][c] | 0.297 | – | 0.309 | 0.198 | – | 0.159 | 0.189 | 0.287 |
| 5 | Binomial | Excess | 10% | Log–logistic | 3.2 | 3.2 | 3.2 | 1.84 | 1.84 | 1.63 | 2.1 | 1.42 |
| | | | | Log-normal | 3.31 | 3.31 | 3.31 | 1.98 | 1.98 | 1.78 | 2.23 | 1.58 |

**Notes.**

[a]Starting values required to get the reported results.

[b]Potential problem with PROAST reporting the same values for the Hill and the exponential model. In the EFSA report results for Hill were: BMD=0.302 and BMDL=0.205.

[c]Not possible to fit a three-parameter Hill model for these data in BMDS. In *bmd* a three-parameter log-logistic model was used instead.

been shown that they result in a lack of power compared to statistically sound alternatives, such as using the Poisson or negative binomial distribution (*Stroup, 2013*; *Stroup, 2015*).

The package *bmd* provides BMD estimation for practically all dose–response functions available in BMDS and PROAST. The package also allows the use of fractional polynomials, particularly suitable for model averaging (*Faes et al., 2007*; *Namata et al., 2008*), and several hormesis models, as well as semi-parametric alternatives (*Piegorsch et al., 2014*; *Lin, Piegorsch & Bhattacharya, 2015*). An additional advantage of *bmd* is the possibility to estimate BMDL based on sandwich variance–covariance estimates that provide robust standard errors, thus making allowance for model misspecification related to the distributional assumptions. This option is not available in any other specialized BMD software packages. Several different types of bootstrap can be used to estimate BMDL, and several options for estimating BMD and BMDL, including model averaging, are available. Also, other available or future software extensions of **R** can be directly used together with *bmd*.

When comparing results from *bmd*, BMDS, and PROAST, we mostly observed agreement between estimated BMD and BMDL values. However, we also found some differences. We found small differences that most likely were caused by different underlying optimization algorithms. In some scenarios, we also found larger differences. Differences between *bmd*/BMDS on one side and PROAST on the other side were most likely due to the fact that PROAST assumed a log-normal distribution whereas BMDS/*bmd* assumed normal distributions. Depending on the scale for the response, these two assumptions may lead to fairly different results. Moreover, there could also easily be differences due to the different methods used for obtaining confidence intervals and hence BMDL estimates.

At present, *bmd* can only handle models describing a single dose–response curve. However, a future update will extend the built-in functions to handle models for multiple dose–response curves. It would be interesting to include an approach for BMDL estimation for model-averaged BMD where correlations between BMD estimates from the different candidate models would be accounted for (*Jensen & Ritz, 2015*). Likewise, it would be interesting to look at other approaches or models for addressing non-constant variance than assuming log-normally distributed data. Finally, the profile likelihood approach for finding BMDL has been found to perform better than the delta-method–based Wald confidence intervals (*Moerbeek, Piersma & Slob, 2004*; *US Environmental Protection Agency, 2012*). This approach is currently not available in *bmd* but is planned to be part of a future update.

'Defining BMD for a predefined BMR' illustrates that the BMD directly depends on its formulation, which needs to be clearly defined and may thus be perceived as superficially more technically involved than statistical testing. Thus, the BMD approach may be considered a statistical black box, which may affect its use in practice. However, it is accepted or even preferred to the NOAEL approach due to its benefits, as laid out in the introduction. Further, statistical testing itself is scrutinized when used as a naïve binary decision criterion (e.g., *Wasserstein, Schirm & Lazar, 2019*; *Wasserstein & Lazar, 2016*) and with respect to toxicology (*Kluxen & Hothorn, 2020*; *Hothorn & Pirow, 2020*; *Kluxen, 2020*).

## CONCLUSION

We have demonstrated that the **R** extension package *bmd* allows flexible and straightforward BMD estimation for a broad range of applications in ecotoxicology. Comparisons to existing software showed mostly good agreement between estimates, but also some non-negligible differences due to different statistical methods being used. However, we observed differences between all three software programs: *bmd*, BMDS, and PROAST. The freely available **R** software can be viewed as a reliable and user-friendly alternative for dose–response modelling, which also covers all other conceivable practical statistical requirements for (eco)toxicologists, including BMD analyses.

## ACKNOWLEDGEMENTS

We would like to thank the three reviewers for their helpful and detailed comments and many practical suggestions.

### Funding

This work was supported by a grant from the Danish Environmental Protection Agency (j. no. MST-667-00174). The funders had no role in study design, data collection and analysis, decision to publish, or preparation of the manuscript.

### Grant Disclosures

The following grant information was disclosed by the authors:
Danish Environmental Protection Agency: j. no. MST-667-00174.

### Competing Interests

Felix M. Kluxen is employed by ADAMA Deutschland GmbH. Signe M. Jensen, Nina Cedergreen and Christian Ritz declare that they have no competing interests.

### Author Contributions

- Signe M. Jensen conceived and designed the experiments, performed the experiments, analyzed the data, prepared figures and/or tables, authored or reviewed drafts of the paper, and approved the final draft.
- Felix M. Kluxen and Jens C. Streibig analyzed the data, authored or reviewed drafts of the paper, and approved the final draft.
- Nina Cedergreen performed the experiments, authored or reviewed drafts of the paper, and approved the final draft.
- Christian Ritz conceived and designed the experiments, analyzed the data, authored or reviewed drafts of the paper, and approved the final draft.

### Data Availability

R code and data used in the examples are available in the Supplemental Files.
Source code is available at GitHub: https://github.com/DoseResponse/bmd.

## Supplemental Information

Supplemental information for this article can be found online at http://dx.doi.org/10.7717/peerj.10557#supplemental-information.

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
