# Peer review of "bmd: an R package for benchmark dose estimation"

_PeerJ, doi:10.7717/peerj.10557_

## Round 0.1 · original submission · Major Revisions

Please carefully address the Reviewers' concerns and comments, paying special attention to including more background information for (eco)toxicologists about BMD estimation as well as stating clearly the advantages of the proposed R-package compared to the existing software for BMD modelling.

Reviewer 1 ·

Basic reporting

No comments

Experimental design

No comments

Validity of the findings

The scientific community should be encouraged to develop novel tools for BMD modeling, considering the drawbacks of the current software such as US EPA’s BMDS and Netherland’s PROAST. This "bmd" R package developed by Jensen et al. has some advantages over the BMDS/PROAST. Notably, one of the major claimed novelty of "bmd" is on the model averaging. However, as far as I know, BMDS has already incorporated the function of Bayesian model averaging for dichotomous data, and the development of model averaging for continuous data is ongoing and proposed to be done this year. In addition, the BBMD system (Shao et al. 2018) have already developed model-averaging BMD estimations for both dichotomous and continuous data.

I do appreciate the authors efforts of developing alternatives to the current BMD tools. However, when the authors stated that this R package “can be seen as a flexible alternative to BMDS and PROAST”, they need to make clear arguments on when or under what circumstances the risk assessors should use their “alternative”. After all, the two commonly used tools have been extensively validated in the past decade and have good credibility. This brings out another issue – only several datasets were analysis and there were no parallel comparisons between the results from this R package and those from BMDS/PROAST. Is there any difference on the value of BMD/BMDL estimation? If so, how would you explain the differences? These questions cannot be simply ignored especially for a methodology paper. Besides, the use of “bmd” R package had already been mentioned in the previous paper five years ago (Ritz et al. 2015), which makes me wonder the purpose of the current manuscript. I thought there would be a relatively large-scale comparison between BMD estimates using “bmd” vs BMDS/PROAST, basically to show the validity of “bmd”. Otherwise, how would you justify the validity of these results?

When compared to the BMDS, the best argument of this R package could be the flexibility of handling the count data, the method of which has been mentioned in the parent R package (Ritz et al. 2015). In the current manuscript, using just one dataset is apparently not enough to justify the validity of BMD estimation for count data using this new package. Besides, how does the results compare to those from BMDS, when the count data were treated continuous? The authors were correct that this package may have potentials in the field of ecotoxicology. As far as I know, the count data in other fields of toxicology is not common – most of the endpoints are dichotomous or continuous. Therefore, focusing on “selling” the package using much more datasets (especially count data) on ecotoxicology can be helpful.

Reviewer 2 ·

Basic reporting

The authors don’t seem to have a clear motivation or audience for this manuscript, which I think has led to its disorganization and lack of clarity. What is the authors’ motivation for writing this package and manuscript? How is the package novel, useful, and important? Further, to whom are the authors targeting this manuscript – is it to toxicologists who would use bmd to estimate BMRs and BMDs, to statisticians/mathematicians interested in the mathematical basis of this package, or to some other group altogether? Selfishly as a toxicologist, I would think this paper would be most widely read by toxicologists wanting to use this package for BMD estimation. If so, the authors need to rewrite the manuscript with that lens. The authors assume too much background understanding of BMDs, dose-response, risk assessment, and other toxicological ideas that is not appropriate for a non-tox focused journal like PeerJ. The authors also assume that the reader understands the different statistical structures outlined in Section 2 and why they are different for different data structures. The authors also are not careful enough with their word choice, especially in regard to chemical concentrations vs organismal response, which is crucial for a discussion of a metric that describes organismal response to different chemical concentrations (doses). The writing needs a lot of reworking with this consideration – I’ve pointed out a few specific places under “general comments” below in which I thought it was unclear, but this needs to be considered throughout the manuscript.
Further, I am not overly familiar with PeerJ’s “Bioinformatics and genomics” section, but I do not completely agree that this is relevant to that section. It is description of an R package for the estimation of a statistical metric specific to toxicology describing a biological response, but this isn’t the type of analysis when I think of when I think about “bioinformatics,” which usually is in relation to large data sets or high-throughput processing.

Experimental design

The authors do not clearly define their research question or the knowledge gap they seek to fill. I do think the examples portion of this manuscript is the strongest part of it, although the authors need to be more explicit in how they chose certain analytical steps (e.g. choice of risk definition and BMR).

Validity of the findings

The authors do not clearly define the impact or novelty of this work. The authors’ conclusions in the “Conclusion” are not supported by the manuscript (see my specific note below).
I think this is the section to report on the provided data and code – overall, the data and R code are provided in a way that is very clear and easy to use. I particularly appreciate the Word doc which carefully goes through the R code for all of the examples – this is excellent. The only part of the code that didn’t work for me is the sandwich estimation of the CopperTemp.m2 model – the block of code on the top of pg 5 gave me this error:
Error in UseMethod("estfun") :
no applicable method for 'estfun' applied to an object of class "drc"
I have two more questions about the R code/analysis: first, for the bootstrapped estimates for example 3.3 on pages 11 and 12 of the word doc, I got the exact same BMD and BMDL estimates as the example code. If these are bootstrap techniques, shouldn’t they be stochastic? I’m probably missing something here because I’m not a statistician, but I wasn’t expecting to get the exact same results, down to four decimal places, as the example code.
Second, what is the significance/meaning of a negative BMDL estimate (pg 13 the BMDL estimate for alpha.m2.bmd)? How would one use this negative BMDL in practice?

Additional comments

1. The first sentence of the abstract makes some assumptions that the reader knows what the authors mean by “dose”, “hazard characterization”, and “risk assessment”. Nowhere in the abstract do the authors make it clear that this package is designed for use in toxicology and chemical risk assessment (or however the authors wish to describe it). Suggest revision to make that clear since PeerJ is not a specialized tox journal.
2. Introduction, L16-23: I’m a toxicologist myself and I don’t understand this introduction to BMD. How is this different from more commonly calculated tox statistics such as NOEC or LOEC (this seems almost identical to a NOEC to me)? More specifically, what “predefined small change” is used? Are there established, normal changes (5%? 50%?) that are usually used for BMD estimation? By “background level”, do the authors mean control response or the background concentration of the contaminant? How is “experimental design” directly included by calculating a confidence interval (L22-23)? And the BMD refers to the contaminant concentration, but this is not clear when the authors talk about “reference values” and “limit values”. Overall, this important introductory section needs to be rewritten with careful word choices to make it very clear what BMDs are, what they estimate, and what they’re used for. Be explicit if the changes or limits or values are in regard to organismal response or chemical concentration. Figure 1 is not at all helpful in this introductory section (e.g. what is p0?).
3. Introduction, L44-45: The authors do a good job of reviewing the software available for BMD estimation but only state that the package they’re describing here is a “flexible alternative”. What specifically was the goal of developing this package, if there are already software packages that are freely available (I think?) online? How is bmd different from what’s already available?
4. L51-59: Again, this section is very confusing to me, specifically L52-53 – how could there be absolute and relative changes when you’re comparing the response to the “background level”? Should the responses always be considered in comparison to the control (so relative)? Do the authors mean that comparisons of the response to prior knowledge are “absolute” (as they describe in L66-67)? Further, the authors use the parameter p0 for the first time in L59 as an example of the opposite to the standard case before they’ve ever defined p0 or adequately described how a standard case (BMD for an increasing dose-response relationship) is estimated. To me, the authors are making too many assumptions on the background knowledge of the reader and should rewrite this section and the introduction to be clearer.
5. L66: “mean background level/risk of an adverse effect” – does p0 refer to the chemical concentration or the organismal response? Looking at Figure 1, I think it refers to organismal response but this explanation is confusing.
6. L71: what do the authors mean by “risk definition”? It should be explained at the beginning of the methods section. And I don’t understand what the reader is supposed to glean from section 2.1 – how would I, as a toxicologist wanting to use this package, use this information to decide how to use bmd? Outside of the different data types, how do I decide which “definition” to use?
7. L78-86: I don’t understand this – why isn’t p0 just the control response and is instead estimated assuming a normal distribution? The parameter x0 is the “level for which the response is considered adverse” – is this the level of the chemical (concentration) or organismal response?
8. L205: define BIC
9. The examples are overall very helpful. The authors chose very different examples of various exposure types (soil vs aquatic), organisms, endpoints, and issues with the data (e.g. overdispersion in example 3.2) that strengthen this paper and the reader’s ability to use these examples for his or her own work. It would be helpful if the authors explain how they chose the “definition” for each example (e.g. L252 the authors chose the “additional risk definition” for this example – why?). Similarly, how do the authors chose the BMR? It is 0.05 for example 3.1 (L256) and 0.1 for example 3.2 (L272) – why?
10. L252-260: Similar to my comment for Figure 1, what do the units mg/kg refer to in example 3.1? I think it’s mg chloroacetamide/kg soil, assuming the earthworms were exposed through the soil, but this should be explicitly defined.
11. L285-6: what is the evidence for overdispersion? How would one know that there is “more variation than can be explained by the model”? The authors explain an indication of overdispersion in L294 but that’s only after it has been corrected for.
12. L328: “do” should be “due”
13. L349: “analyzes” should be “analyses”
14. L352-363: a summary of this should be included in the introduction to describe the differences between bmd and existing programs (see my earlier note). The authors need to include in this section and others the significance of these differences, however. From my reading, the main difference between bmd and BMDS and PROAST is that it’s connected to drc, includes fractional polynomials, semi-parametric alternatives, and standard error estimations. The authors do a good job explaining the significance and utility of standard error estimations in L359-360, but the significance of the other differences are not stated.
15. L386-388: the authors conclude by saying that the “BMD methodology has great potential in ecotoxicology” and it is applicable to “various analytical issues” but they have not described that explicitly in any other part of this paper. I cannot make those conclusions having read this paper, and the authors need to more clearly lay out their evidence for these claims throughout the paper.
16. Figure 1 legend: define BMD, BMDL, BMR, and p0 in the figure legend, along with what the units of mg and kg refer to (I assume mg is of the contaminant and kg is soil because it was a soil exposure?).

·

Basic reporting

- Many of the math expressions in the manuscript are improperly formatted. For example, I have no idea what is going on in equations (7) (8) and (9). Equations/expressions need to be cleaned up throughout the manuscript.

- Line 52 - 55. This text needs to be rewritten and clarified. In reality, some fo the definitions are only applicable to certain data types. That is the meaning is relative to the type of data. For count, the meaning is usually based upon the rate etc. This needs to be clarified.

-Their Hybrid response is based upon the normal distribution with constant variance. It would nice to know if their approach extends to non-constant variance or Log-normal models. The latter two are usually used in Europe and North-America.

- It is unclear what the author's mean by 'after-fitting.' on line 120. This is the first time I have encountered this term in the literature (statistics or otherwise). Some discription should be given to the reader.

Experimental design

- There is no comparison to the other software packages (e.g. BMDS and PROAST). As many of the models are shared between all three packages some attempt at comparison should be made.

- Additionally timings of the algorithms should be given. Choice between the use of R-BMD bootstrap model averaging and PROAST bootstrap model averaging may come down to computational efficiency. Currently the online EFSA/ PROAST tool takes 10s of minutes for several hundred bootstraps. It would be nice to know the difference.

- The Author's do not compare their estimates using the variety of methods they propose. For example, table 3 gives bootstrap BMDL estimates. They should give a table that gives the results for every method presented.

Validity of the findings

- The findings are sound.

Additional comments

The paper is very well written and should be ready for publication given correction of the above concerns.

---

## Round 0.2 · Minor Revisions

Thank you for revising your manuscript. However, additional revisions are needed prior to the acceptance for publication. Please address the comments of Reviewer 2 and resubmit your manuscript.

Reviewer 1 ·

Basic reporting

No more comments

Experimental design

No more comments

Validity of the findings

No more comments

Reviewer 2 ·

Basic reporting

I am re-reviewing this manuscript after submitting reviews on the initial submission. Overall, the authors did a good and thorough job responding to my comments and addressing them through changes in the manuscript. I appreciate the authors hard work and time on this. There are a few points that are still unresolved or unclear which I’ve listed under ”General comments”, but overall I think this manuscript is very close to submission with minor changes (mostly focused on the clarity and structure of significantly rewritten sections such as the Introduction and Validation of the R Package).

Experimental design

The authors responded to my comments sufficiently and added a new section comparing the bmd package to the existing programs (Section 4) which is quite good (although needs editing and restructuring, see General Comments below). Overall, the experimental design is strong.

Validity of the findings

The authors responded to my comments sufficiently, however a new paragraph in the discussion is concerning (L627-633) and seems to potentially undermine their conclusions and the utility of this package. The authors need to address this.

Additional comments

L15-29: I appreciate the authors taking time to rewrite this section for clarity and it is improved but is still a bit confusing. What do the authors mean by ”agent” in L16? Why not use the word chemical or stressor? And the authors need to define the many acronyms they use in L24-25. This whole paragraph needs to be closely re-read for mistakes and edited for clarity (e.g the last sentence doesn’t make sense: ”also when exposure is a concentration or a contaminant level”?)
Overall, the introduction is much improved and does a good job explaining the background and motivation for the paper. However, there are typos, confusing language, fragmented or run-on sentences, and awkward or non-existent transitions between paragraphs throughout that will lead to a lack of clarity (e.g. L51 – what do the authors mean by ”compare e.g. Bresica, 2020”? Is that reference an example of BMD or NOAEL or the differences between them?). The introduction needs to be closely re-read by all the authors.
L157: the authors did not resolve my comment below – they literally rearranged the words that were already there and it is still unclear.
Previous comment and authors’ response:
Reviewer: The parameter x0 is the “level for which the response is considered adverse” – is this the level of the chemical (concentration) or organismal response?
Authors: Thank you for the comment. This has been changed to: x_0 is the response level considered to be adverse.
L257-274: I again appreciate the hard work the authors have put into this revision, including this new section which I think does help the reader substantially. However, like the introduction, there are small typos and errors throughout that need to be addressed with careful re-reading by all co-authors (e.g. L263 ”cause” should be ”course” and the sentence that runs L270-272 doesn’t make sense).
”Validation of the R Package”: I really like this idea behind this section and I think it does a great job of comparing bmd to the existing packages! However, again, this section needs to be closely re-read by all authors (the sentence that runs L512-513 does not make sense in professional English). Also, the structure of this section is all over the place – the authors present results (L536) and then return to methods (L537-561) and then present more results (L562) before discussing the results (starting in L563). I think this section needs to be reorganized.
L627-633: this paragraph is concerning – the authors do a good job discussing differences in results between the packages in L566-581 but here the differences sound much more significant. I also don’t understand what the authors mean that ”none of the packages can be considered the ”truth”” (L631). The authors need to be much more clear on this important point, since the implications of differences between bmd and BMDS and PROAST that have been ”validated extensively” (L627) are potentially a big problem for users interested in bmd.

---

## Round 0.3 · accepted · Accept

I have no further comments.